# Endangered Nectar-Feeding Bat Detected by Environmental DNA on Flowers

**DOI:** 10.3390/ani12223075

**Published:** 2022-11-08

**Authors:** Faith M. Walker, Daniel E. Sanchez, Emma M. Froehlich, Emma L. Federman, Jacque A. Lyman, Meagan Owens, Kristen Lear

**Affiliations:** 1Bat Ecology & Genetics Lab, School of Forestry, Northern Arizona University, Flagstaff, AZ 86011, USA; 2Pathogen and Microbiome Institute, Northern Arizona University, Flagstaff, AZ 86011, USA; 3Integrative Conservation and Forestry & Natural Resources, University of Georgia, Athens, GA 30602, USA; 4Bat Conservation International, Austin, TX 78746, USA

**Keywords:** environmental DNA, eDNA, Chiroptera, pollination, high-throughput nucleotide sequencing, DNA metabarcoding

## Abstract

**Simple Summary:**

Nectar-feeding bats may leave DNA behind on flowers and this DNA may be detectible with genetic tools. Determining whether this is the case is important because some of these bat species follow “nectar corridors” during their migrations, and these corridors should be located for conservation and management. We collected flower samples from agaves that were visited by the Mexican long-nosed bat and developed two eDNA detection methods (DNA metabarcoding and qPCR) to assess whether bat DNA could be detected. We found that both methods were highly successful in detecting this bat species and other mammals and arthropods that may interact with agaves. We suggest that, together with a further proof of concept, these detection methods can be used for identifying nectar corridors and foraging grounds for the Mexican long-nosed bat.

**Abstract:**

*Leptonycteris nivalis* (the Mexican long-nosed bat) is an endangered nectar-feeding bat species that follows “nectar corridors” as it migrates from Mexico to the southwestern United States. Locating these nectar corridors is key to their conservation and may be possible using environmental DNA (eDNA) from these bats. Hence, we developed and tested DNA metabarcoding and qPCR eDNA assays to determine whether *L. nivalis* could be detected by sampling the agave flowers on which it feeds. We sampled plants with known bat visitations in the Sierra Madre Oriental in Laguna de Sanchez (LS), Nuevo León, Mexico, and in the Chisos Mountains in Big Bend National Park, TX, USA (CB). A total of 13 samples included both swabs of agave umbels and cuttings of individual flowers. DNA metabarcoding was performed as a PCR multiplex that targeted bats (SFF-COI), arthropods (ANML-COI), and plants (ITS2 and rbcL). We targeted arthropods and plants in parallel with bats because future metabarcoding studies may wish to examine all the pollinators and plants within the nectar corridor. We developed and tested the sensitivity and specificity of two qPCR assays. We found that both DNA metabarcoding and qPCR were highly successful at detecting *L. nivalis* (11 of 13 for DNA metabarcoding and 12 of 13 for qPCR). Swabs and flower cuttings and both qPCR assays detected the species over four replicates. We suggest that *L. nivalis* leaves substantial DNA behind as it forages for nectar. We also suggest that future studies examine the time since sampling to determine its effect on detection success. The DNA metabarcoding multiplex will be useful for parallel questions regarding pollination ecology, while, with further testing, the qPCR assays will be effective for large-scale sampling for the detection of migration corridors and foraging areas. This work may be relevant to other nectar-feeding bat species, which can likely be detected with similar methodologies.

## 1. Introduction

A promising field of study that is a recent spinoff of non-invasive genetics involves environmental DNA (eDNA), which continues to extend into novel applications. eDNA is genetic material collected from an environmental sample without any efforts taken to isolate the organism itself [1]. Such samples can include water, air, or sediment and can involve whole cells, extracellular DNA, or, in the case of microorganisms, whole organisms [2]. For example, DNA has been captured from river otters using water [3] and community members using spider webs [4]; such studies often have strong applications to conservation [5,6].

The eDNA arena has been overwhelmingly focused on aquatic systems, with gold standard-sampling and quality control methods already in place [7]. Terrestrial applications have received less attention [8], and most studies have involved water as a sampling source and have largely included species that are common or invasive (e.g., [9,10]). The utility of eDNA is due to its assay sensitivity, and there is great potential to detect rare or endangered terrestrial species with creative sampling methods [11]. Wildflowers have been used to sample the eDNA of terrestrial arthropod pollinators and predators [12], and hence may be useful for capturing DNA from vertebrate pollinators.

*Leptonycteris nivalis* (the Mexican long-nosed bat) is a migratory, nectar-feeding species found in Mexico and the southwestern United States that would benefit from eDNA detection via salivary cells left on agave flowers. The species is listed as endangered in the U.S. [13] and by the IUCN Red List of Threatened Species [14,15]. The species is also considered threatened in Mexico [16]. *L. nivalis* consumes nectar and pollen from at least 49 flowering plant species across its migratory range between central Mexico and the southwestern United States [17]. Each year, females undergo a migration of over 1200 km between their mating roosts in central Mexico and their maternity roosts in northern Mexico and the southwestern United States. During migration, the bats follow a “nectar corridor” of flowering agave plants (*Agave* spp.) and cacti (Family *Cactaceae*) [18,19,20] but rely primarily on agave nectar in the northern portion of their range [21,22,23]. However, the loss and degradation of agaves and other food resources near their roosting sites and along migratory corridors is thought to be one of the main threats to the species [17]. The IUCN has stated that one of the critical conservation needs for this species is the protection of their foraging habitat [15], and the U.S. Fish and Wildlife Service identifies this as Recovery Task 2.2 in the Species Recovery Plan [24]. Current conservation efforts for the species are focusing on increasing the availability of flowering agaves around key roosting sites and along migratory corridors [25]. However, the current migratory corridors remain unknown. The corridor between Emory Cave (a maternity roost in Texas) and the Big Hatchet Roost (a late summer transition roost in New Mexico) is especially important, as this is the last critical section of the migratory route and climate change predictions show that this region may play a more significant role for *L. nivalis* in the future [26,27]. Identifying the migratory corridors and foraging grounds will allow for the targeted protection and restoration of the foraging resources in these areas.

The traditional methods for surveying for the presence of nectar bats (e.g., mist netting, acoustic monitoring, and the camera-based monitoring of bat visits to foraging resources) often prove expensive, time-intensive, and unreliable. Identifying the foraging areas of *L. nivalis* is also complicated by the fact they can forage over 50 km from their roost each night. The use of GPS transmitters is difficult given that the bats are under the weight requirement for GPS tags that automatically upload location data, and the recovery of archival tags is often difficult if not impossible. In addition, because of the difficulty in collecting high-quality echolocation calls from *Leptonycteris* in field settings [28], the use of acoustic techniques to distinguish between *L. nivalis* and its sister species, the Lesser long-nosed bat (*Leptonycteris yerbabuenae*), which co-occurs with *L. nivalis* in some parts of their ranges, is problematic [15,29]. The collection and analysis of bat eDNA from forage plants such as agaves may be an efficient, cost-effective technique for detecting nectar-feeding species such as *L. nivalis*.

There are two methods of eDNA detection: eDNA metabarcoding and eDNA via quantitative PCR (qPCR). In this context, DNA metabarcoding uses universal polymerase chain reaction (PCR) primers on mixed DNA samples in a high-throughput, next-generation sequencing framework in order to identify one or more species that have interacted with a sample. The simultaneous targeting of species means that it is possible to monitor diverse members of a community by applying multiple markers and sequencing techniques in parallel. qPCR, on the other hand, involves the development of primers that are specific to the target species, and hence essentially answers a yes/no question about whether the DNA of the species is present or not. While it does not return community information, the advantage of qPCR is that, after development, it is sensitive and inexpensive to screen samples.

Here, we aim to build upon existing methods and approaches to move forward the eDNA field for terrestrial species. We (1) take a first step toward a standardized method for the detection of *L. nivalis*, and (2) test the feasibility of the use of a multiplex of primers to simultaneously identify bat–arthropod–plant communities. We apply DNA metabarcoding and qPCR assays and compare the detection efficacy of both approaches. The relative performance of these assays on different sample types will determine the most effective means to detect *L. nivalis*. We apply a multiplex of markers to detect *L. nivalis* from its salivary cells deposited on agave flowers, while illustrating that arthropod and plant species can be simultaneously detected. We posit that the detection of all three taxonomic groups will be important to identifying nectar corridors, the plants they contain, and their pollinators and non-pollinators. This work shows that the detection of a bat pollinator from eDNA on flowers can be highly successful and may be so for other bat and bird pollinators as well.

## 2. Materials and Methods

### 2.1. Ethics Statement and Permits

This study was approved by the Institutional Animal Care and Use Committee (IACUC) of the University of Georgia (Permit A2015 03-011-Y1-A0), the United States Department of the Interior National Park Service Big Bend Scientific Research and Collecting Permit #BIBE-2021-SCI-0020, and a Texas Parks and Wildlife Scientific Research Permit. No bats suffered injury or mortality as part of this study.

### 2.2. Field Methods

We collected a total of 13 samples for eDNA analysis from two agave plants with known visits from *L. nivalis*. We collected samples from one plant in the Sierra Madre Oriental in Laguna de Sanchez (LS), Nuevo León, Mexico (14 July 2018; *n* = 4), and one plant in the Chisos Mountains (CB) in Big Bend National Park, Texas, USA (25 and 26 July 2021; *n* = 9). We recorded nightly *L. nivalis* foraging activity with a digital video camera placed approximately 10 m away from the focal agave (Sony FDR-AX33 and FDR-AX53, using the Nightshot feature). We used two infrared lamps (IR6 Lamps, Wildlife Engineering, Tucson, AZ, USA) to provide supplemental infrared lighting such that all umbels (flower clusters) with open flowers were clearly illuminated. While watching the bats visiting the flowers, we noted the location of each visit on the flower umbels. After completion of camera monitoring, we either sampled in the early morning before leaving the field site (*n* = 7) or at approximately 18:00 h (*n* = 6). Samples were collected by either removing an individual flower from a visit site on the umbel (cuttings, *n* = 10; Figure 1) or swabbing the entire visited umbel using a polyester swab (swabs, *n* = 3) (Figure 2). Collection methods for each were standardized. For cuttings, we used garden shears or a telescoping pruner and a net to cut down one umbel from a given plant. Shears and pruners were decontaminated (10% bleach followed by flame sterilizing with 95% EtOH in indoor setting) between samples and a plastic bag was used to line the net during collection. The individual flowers were selected from portions of the umbel that had received *L. nivalis* visits throughout the night. Different scissors (decontaminated after previous nights) were used for each of the Big Bend cuttings. For swabs, we used a polyester tip applicator (Puritan SKU# 25-806) attached to a long pole to swab one entire umbel that had received *L. nivalis* visits (Figure 2). A new plastic bag was placed around the head of the pole between samples to prevent cross-contamination. Cuttings were stored in 15 mL conicals of RNAlater (*n* = 3; Figure 1) or unpreserved in re-sealable plastic bags (*n* = 3). Swabs were stored in 2 mL vials of RNAlater. Samples were stored in a cooler in the field and then in a −20 °C freezer prior to DNA extraction.

### 2.3. DNA Extraction

We targeted the stigma, style, and the ovaries of flower cuttings—locations where bats would make contact with their tongue and face during feeding. We extracted DNA from flower cuttings (*n* = 10) or umbel swabs (*n* = 3) with a Qiagen DNeasy Blood and Tissue Kit (Qiagen, Carlsbad, CA, USA) using a modified protocol for buccal swabs described in Walker, et al. [30]. Volumes of Buffer ATL and proteinase K were doubled so as to submerge the agave cuttings. We extracted DNA from cuttings using four variations. The first variation (Cut 1) was DNA extraction from half of a stigma, style, and ovary that were shipped unpreserved in re-sealable plastic bags (*n* = 4 from LS; *n* = 3 from CB). The second variation (Cut 2) was the same DNA extraction method but from flowers that were immediately stabilized in RNAlater contained in 15 mL vials (*n* = 3). From the same 15 mL vials used for Cut 2, the third variation (Cut 3) was a DNA extraction of the other half of the flower cuttings and included the remaining RNAlater solution (*n* = 3). The fourth variation was similar to Cut 1 but involved a DNeasy Plant Mini Kit (Qiagen, Carlsbad, CA, USA) (*n* = 4), and was conducted on the same four samples from the LS locality in Cut 1. Prior to using the plant kit, samples were homogenized in lysis buffer with a 5/32-inch stainless-steel grinding ball for 5 min at 1500 rpm via a 1600 MiniG Homogenizer (SPEX Sample Prep, Metuchen, NJ, USA). 

### 2.4. DNA Metabarcoding

#### 2.4.1. Library Preparation

Using existing group-specific primer sets (Table 1), we performed PCR-amplification in a multiplex reaction to simultaneously target bats, plants, and arthropods. We used the SFF-COI primer set for bats [30], the ANML-COI primer set for arthropods [31], and two primer sets for plants, targeting either rbcL or ITS2 [32,33]. The rbcL primer set targeted short fragments to detect degraded plant DNA, whereas the ITS2 primer set targeted longer fragments to allow for greater taxonomic resolution. All primers were modified with 5′ universal tails (Table 1) for two-step library preparation [34]. In the first step, target regions are PCR amplified, which incorporates the universal tail into the amplicon. In the second PCR step, primers containing a unique index and Illumina adapters bind to and extend from the universal tail to render the amplicon flow-cell ready. 

For the first PCR step, reactions were carried out in 25 μL volumes containing 1× KAPA2G Fast Multiplex Mix (MilliporeSigma, St. Louis, MO, USA), 0.1 μM each ITS2 primer, 0.3 μM each rbcL primer, 0.4 μM each SFF-COI primer, 0.2 μM each ANML-COI primer, and 2 μL undiluted DNA template, with PCR-grade water making up the remaining reaction volume. PCR amplifications were performed with a SimpliAmp Thermal Cycler (Applied Biosystems, Thermo Fisher, Waltham, MA, USA). Thermal cycling conditions were as follows: 3 min denaturation at 95 °C; 5 cycles of 95 °C for 15 s, 45 °C for 30 s, and 72 °C for 30 s; then, 35 cycles of 95 °C for 15 s, 56 °C for 30 s, and 72 °C for 30 s. This was followed by a final extension cycle at 72 °C for 10 min. Included on each PCR plate was a negative template control (NTC) with PCR-grade water substituted for DNA template. As a positive control, we included a bat mock community containing pooled genomic DNA of *L. nivalis*, *Eptesicus fuscus*, *Eumops perotis*, *Lasionycteris noctivagans*, *Lasiurus cinereus*, *Myotis occultus*, *Nyctinomops macrotis*, *Tadarida brasiliensis*, and *Euderma maculatum*. We also included a plant mock community containing the pooled DNA of *Bromus carinatus* var. *marginatus*, *Festuca arizonica*, *Gnaphalium exifolium*, *Houstonia wrightii*, *Juncus saximontanus*, and *Oxalis dillenii.*


To determine whether the multiplex PCR affected detection of *L. nivalis* or other bats, we amplified the samples from the Big Bend locality, as well as the same mock community, using only the SFF-COI primer set. We amplified the SFF-COI primer set in a 15 µL reaction with 3 µL of DNA template, 8.46 µL of PCR-grade water, 1.5 µL of 10× Mg-free PCR buffer (Invitrogen, Thermo Fisher Scientific, Waltham, MA, USA), 1.5 mM MgCl_2_, 0.2 mM each dNTP, 0.2 µM each primer, 0.16 µg/µL of bovine serum albumin (Ambion Ultrapure BSA), and 0.03 U/µL of PlatinumTaq DNA polymerase (Invitrogen, Thermo Fisher Scientific). Thermocycling involved an initial denaturation of 94 °C for 5 min, followed by 5 cycles of 94 °C for 1 min, 45 °C for 1.5 min, and 72 °C for 1 min. This was followed by 35 cycles of 94 °C for 1 min, 60 °C for 1.5 min, and 72 °C for 1 min, concluding with final extension step of 72 °C for 10 min.

We prepared flow-cell ready amplicon in a second PCR step. Libraries were amplified in 25 µL reaction volumes with purified 2 µL amplicon from the previous PCR step, 12.5 µL 2× Kapa HiFi HotStart ReadyMix (Roche Sequencing, Wilmington, MA, USA), 8.5 µL PCR-grade water, and 1 µL each index primer (starting concentration: 10 µM). Thermocycling included an initial denaturation of 98 °C for 2 min, 8 cycles of 98 °C for 30 s, 60 °C for 20 s, and 72 °C for 5 min, concluding with a final extension of 72 °C for 5 min. Libraries were pooled and run on Illumina MiSeq 600 cycle kits (Illumina, San Diego, CA, USA) among two runs. The first run was a pilot analysis in 2018 containing only the samples from the Mexico locality (LS). The second sequencing run contained samples from the Chiso Basin locality (CB) and was sequenced in 2022.

#### 2.4.2. Sequence Processing and Taxonomic Classification

We separated markers and removed priming regions using cutadapt v4.0 in paired-end mode [35]. We subsequently pre-processed each marker individually using QIIME2 v2022.2 [36] and custom Tidyverse [37] scripts in R v4.2.1 [38]. Exact methods and parameters varied for each marker in downstream analysis and they are available in Appendix A. Using DADA2 [39], we truncated reads and filtered by quality, derived amplicon sequence variants (ASV), joined paired-end reads, and removed chimeric sequences. We performed this step individually for reads of each of the two runs because error models trained in DADA2 are run-specific. For simplicity, we refer to ASVs or any post-clustered ASVs as features. We then merged run-specific feature tables for each marker. For the bat marker, we only retained features of 202 bp in length, a consistent insert length among bat species for this marker [30], and then post-clustered them into operational taxonomic units (OTUs) [40]. For plant and arthropod markers, singleton features were removed, post-clustered into OTUs [40,41], and filtered to retain major taxonomic groups matching our group-specific reference libraries [42]. This was performed to exclude non-target taxa that co-amplified with our group-specific markers—such as bacteria, fungi, or nematodes—and prevent false positive classifications. We did so using hidden Markov models [43] or least common ancestor classification (LCA) [44] of BLAST searches [45] against the National Center for Biotechnology Information’s (NCBI) GenBank nt database [46]. We retained features locally aligning to kingdom Viridiplantae for both plant markers and to phylum Arthropoda. Since our goal was to identify the source plant, we retained the most abundant plant features by removing features represented in fewer than 5% of the reads in a sample.

We determined taxonomy using Naïve Bayes classification [47] and by cross-referencing those classifications against species known to occur in our study localities. We classified the features against group-specific reference libraries, which were validated for each marker elsewhere using kmer-based classification [30,42,48,49,50]. For the bat marker, any feature not classified to the species level was re-classified using LCA classification against the GenBank nt database. This was to correct for under-classification or to identify co-amplified, non-bat vertebrates. We attempted to classify plant ITS2 features using the PLANitS reference library [48] but were initially unable to classify any expected agave features at higher resolution than the family-level (Asparagaceae; as labelled in the PLANitS reference library). We found that representative agave sequences in this reference library included homopolymer runs of degenerate (N) characters, which could have affected training and classification. To improve lower-level taxonomic classification, we re-built an ITS2 reference library (see Appendix A for further details) from the Barcode of Life Database [51] and GenBank using the RESCRIPt pipeline [52]. This resulted in 43,154 unique taxa-barcode pairs (phylum Magnoliophyta only). Taxonomies for each marker were examined by cross-referencing species lists for plants and vertebrates [53] (https://irma.nps.gov/NPSpecies/; accessed 26 August 2022), as well as via a priori knowledge of assemblages in the study localities. For plants and mammals, any species-level classification for a taxon not known to occur in the study localities was determined to be over-split. Over-split classifications were subsequently collapsed to genera or listed with potential sympatric congeners if known. Any genus not known to occur in the study area was collapsed to family level. Comprehensive lists of arthropod species were unavailable for our study localities. Instead, we focused our interpretation of arthropod taxa based on features and broader taxonomic descriptions. All taxonomies determined in this study are available in Appendix A.

##### 2.4.3. qPCR Detection

We designed two candidate qPCR assays (LENIS1 and LENIS3, Table 2) using recommended procedures [54]. From NCBI GenBank in June 2021, we downloaded all available cytochrome B (cytb) sequences for *L. nivalis* (*n* = 1) as well as *L. yerbabuenae* (*n* = 77), its most closely related, sympatric species in the region encompassing our study localities. We sequenced the same region for an additional three *L. nivalis* specimens (Angelo State Natural History Collections: ASK11601, ASK116041, and ASK116051). Sequences with degenerate characteristics were removed using screen.seqs function in mother [55]. Those retained were aligned and trimmed to equal length (1067 bp) using MEGA7 [56]. We used PrimerProspector v1.0.1 [57] to find all possible primer or probe sequences (ranging from 18 to 28 bp in length), favoring *L. nivalis* (100% similarity) over *L. yerbabuenae*. We filtered candidate primer and probe sequences based on melt temperature, sequence length, and GC content using open-source python scripts [11], custom R scripts, and the IDT OligoAnalyzer™ Tool (https://idtdna.com/pages/tools/oligoanalyzer; accessed 26 August 2022). We calculated the maximum number of mismatches to non-target sequences (*L. yerbabuenae*) for primers and probes using SequenceMatcher software (https://github.com/dariober/SequenceMatcher/; accessed 26 August 2022). We then screened four promising primer sets against the nr database using Primer-BLAST [58]. We screened against order Chiroptera, *L. yerbabuenae*, *Choernycteris mexicana*, agave, and other vertebrates known to use agave [21,59]. This included wrens (Troglodytinae), woodpeckers (Picoides), hummingbirds (Calypte), orioles (Molothrus), and Finches (Serinus). For *L. nivalis*, we observed no variation in the priming regions, probe regions, or other segments of the amplicon (Appendix A).

Prior to screening unknown samples, we optimized cycling conditions and reagent concentrations, and then evaluated sensitivity and in vitro specificity for each of the two candidate assays [54]. For each qPCR experiment, we used a gBlocks™ (Integrated DNA Technologies) synthetic DNA sequence of *L. nivalis* (based on GenBank accession KC747678.1; 30,568 copies/μL) as a positive control and at least three non-template controls (PCR-grade water in lieu of DNA template). Both assays used the same FAM-labelled, double-quenched hydrolysis probe (3IABkFQ with ZEN quencher, Integrated DNA Technologies). All qPCR reactions were carried out on a QuantStudio™ 7 Pro Real-Time PCR System (ThermoFisher Scientific, Waltham, MA, USA). We first identified optimal annealing temperatures. We prepared 15 µL qPCR reactions with 7.5 µL 2× Environmental Mastermix 2.0 (Life Technologies, Carlsbad, CA, USA), 900 nM each primer, 250 nM hydrolysis probe, and 3 µL gDNA template, with PCR-grade water making up the remaining reaction volume. Conditions included a hot start cycle for 10 min at 95 °C, followed by 45 cycles of denaturing for 15 s at 95 °C, varied annealing temperatures for 30 s (46 °C, 49 °C, 51 °C, 54 °C, 57 °C, and 60 °C on the same run), and a separate extension step for 30 s at 72 °C. We then determined optimal primer and probe concentrations by varying forward and reverse primer concentrations at 100, 300, 600, and 900 nM while holding the probe constant at 250 nM for all 16 combinations [60,61]. Following optimization, we evaluated assay sensitivity and reliability of quantification. The limit of detection (LOD; 95% detection for a single reaction of an unknown) was determined using probit modelling, which also estimates effective LOD for up to eight replicates [54]. The effective LOD was used to inform selection of replicate numbers for screening unknown samples. At 16 replicates per level, we tested a six-level standard curve of fourfold dilutions (*L. nivalis* gBlocks™ KC747678.1), ranging from 3072, 768, 192, 48, 12, and 3 copies per 3 µL gBlocks™ template. LOD, limit of quantification (LOQ), regression coefficients, and metrics (slope, y-intercept, R^2^, and PCR efficiency) were determined from the output of qPCR_LOD_Calc.R (https://doi.org/10.5066/P9GT00GB; accessed on 1 September 2022) with default settings. The appropriateness of the default margin of error for LOQ was also inspected prior to accepting results. We tested the specificity of both assays in vitro with gBlocks™ synthetic DNA (GenBank accession: MH179182, *n* = 1) and genomic DNA (*n* = 2) of three *L. yerbabuenae* specimens; *C. mexicana* (*n* = 1); two sympatric bat species (*Corynorhinus townsendii* and *Myotis lucifugus*) with fewer or equal to 8 mismatches at the priming regions (i.e., identified via Primer-BLAST); and two insectivorous bat species based on known or suspected agave visits, namely, *Antrozous pallidus* [62] and *Euderma maculatum* (K. Lear, personal communication, possible agave visitation).

We screened all agave flower samples in quadruplicate for both assays. Screening included three negative controls, one positive control, and a six-level standard curve of tenfold dilutions (range: 9,170,302–92 copies per 3 µL gBlocks™ template). Reactions were carried out in 15 µL reaction volumes with 7.5 µL of 2× Environmental Mastermix 2.0 (Life Technologies), 250 nM hydrolysis probe, and assay-specific primer concentrations (Table 2). We also included an internal positive control (IPC) in each reaction. We determined inhibition if amplification curves of the IPC were delayed by at least one quantification cycle (Cq) from the non-template controls. Our qPCR reactions included 1.5 µL 10× Internal Positive Control Assay (Life Technologies) and 0.3 µL 50× Internal positive control DNA (Life Technologies). PCR-grade water made up the remainder of the reaction volume. Cycling began with a hot start step for 10 min at 95 °C, followed by assay-specific cycling conditions (Table 2) for 45 cycles. To verify that we amplified the expected amplicon region of *L. nivalis*, we selected eight positive detections from an early screening of the LENIS3 assay and prepared them for next-generation sequencing using the same procedures and software as the metabarcoding methods above. Wells with positive detections were purified using 1× AMPure XP magnetic beads (Beckman Coulter Life Sciences, Indianapolis, Indiana, USA), amplified with UT-modified assay primers, and indexed in a subsequent PCR for Illumina sequencing. Following sequencing, the markers were isolated based on priming regions, dereplicated (ASVs), and classified using LCA classification of BLAST results (described above).

## 3. Results 

*L. nivalis* was successfully detected by DNA metabarcoding in 11 of the 13 samples and by qPCR in 12 of the 13 samples (Table 3). The extraction of DNA from swabs of the flowers performed as well or better than extraction from flower cuttings. The storage conditions performed similarly and would benefit from tests with larger sample sizes. Using the plant kit, three samples showed inhibition whereas all others did not, suggesting that this kit is not reliable for this purpose. The multiplex of different markers detected *L. nivalis* as or more readily as the singleplex (Table 3). Both qPCR assays detected *L. nivalis* (Table 3 and Appendix A), despite LENIS1 being slightly more sensitive (effective LOD of 11 copies per reaction at four replicates; Table 4). We estimated a maximum of 1421 starting copies/reaction for LENIS1 and 1234 copies/reaction for LENIS3; the mean Cq over all samples was 35.56 (SD 0.82). From DNA metabarcoding, we detected four mammal species (6 OTUs), 12 plant genera (16 OTUs from ITS2; 2 from rbcL), and 21 arthropod genera (62 OTUs) (Table 5). All mammal and plant species are known to occur in the study areas. The plants detected were diverse, with ten orders represented. Among the plants, the source plant (*Agave*) constituted the majority of the reads (Appendix A).

## 4. Discussion

We reliably detected *L. nivalis* DNA on agave flowers by both DNA metabarcoding and qPCR, even after the DNA was subjected to a summer day’s heat and sun. The only sample that consistently failed involved a bud that was likely not sufficiently developed enough to attract bats. The high success rate for DNA metabarcoding, which in general, is thought to be less sensitive than qPCR [63], suggests that this bat species leaves a large amount of DNA behind during its flower visitations. Nectar-feeding bats employ various means of transporting nectar to their mouths [64], all of which involve the use of a long tongue that presumably deposits salivary cells on flower parts. The swabs of umbels with known bat visitations were also highly successful at obtaining DNA for species detection, indicating that swabbing may be an effective field survey tool for the detection of these bats while simultaneously being easier to implement on a large scale.

The DNA metabarcoding multiplex performed robustly and without negatively affecting the detection of *L. nivalis*, which means that for a slightly higher cost, one can simultaneously generate arthropod pollinator and plant data. Analogous multiplexes were designed for the identification of species, sex, and diet from the feces of *Gyps* vultures [65], and for species, diet, parasites, and genotypes in bats [66]. The detection of 42 arthropod genera and species corroborates another study showing that arthropod communities can be detected from wildflowers using DNA metabarcoding [12]. The DNA from the 14 non-agave plant genera detected may have arrived on agave flowers through insect or bat pollinators [67] or wind [68]. The detection of deer and foxes is suggestive of airborne DNA, perhaps aided by the umbel being cut and then left on the ground uncovered until sample collection about 18 h later. It is interesting that ringtail cats were detected because this species has been reported to feed on agave nectar [69], including from agaves in both the Chisos Basin and in Laguna de Sanchez on multiple occasions (K. Lear, personal communication). 

At a fraction of the cost of amplifying several markers separately, the DNA metabarcoding multiplex provides the capacity to simultaneously survey plants and arthropods in nectar corridors, which will become increasingly important for managers, researchers, and non-profits as they seek to protect and restore habitats important for *L. nivalis*. For plants, airborne eDNA contains more than pollen [70], and hence provides information about the whole plant community and can be used as a community survey and monitoring method [68]. Likewise, arthropods that are significant for agave species can be detected and their taxonomic and functional groups characterized [12]. The multiplex could also be useful for non-eDNA applications. For example, *L. nivalis* belongs to the species-rich Phyllostomidae family, which exhibits diverse diets of fruits, nectar, insects, vertebrates, and combinations thereof [71,72,73]. Metabarcoding has already been useful in characterizing omnivorous diets in this family with separately amplified markers [74,75] and the multiplex used in our study could potentially be useful in making such an acquisition of dietary data more scalable.

Both qPCR assays detected *L. nivalis*. LENIS1 can detect down to fewer copy numbers while LENIS3 is shorter, both attributes which will become important in scenarios where the time since visitation is unknown and where flowers are subject to daytime heat and sunlight and the potential subsequent degradation of DNA. The qPCR assays will benefit from a larger study that also examines the detection probabilities estimated by statistical modelling and that identifies the influences of environmental variables on detection probability [76]. At that stage, the best performing qPCR assay can be employed on a large scale to identify migratory corridors and foraging grounds; in the meantime, DNA metabarcoding can be used. The use of these assays can also be explored for the detection of nectar bats in potential roost sites, which would aid in the protection of these sites. 

## 5. Conclusions

We found that DNA metabarcoding and qPCR can perform well for detecting *L. nivalis* from agave flowers, given that visitations are known, and the samples are collected either immediately or within 18 h. Before these assays are routinely deployed to identify migratory corridors and foraging grounds of this endangered species, we suggest that further qPCR studies use larger sample sizes to explore the utility of swabs, which are easier to deploy in the field and may be more conservation-friendly since they do not remove foraging resources, compared to flower cuttings. How variables such as duration of time following bat visitation, air temperature, and sun exposure affect the probability of detection should also be explored. Additionally, it is important to determine the probability of occurrence in a setting where bats’ visitation of agave plants is unknown. Ultimately, we suggest that a qPCR assay be used for large-scale screening of an area to determine whether the species is present, and that DNA metabarcoding multiplex be used when it is desirable to additionally understand plant species’ identity, pollination, or plant or arthropod communities. Assays of these types are likely obtainable for many nectar-feeding bats.

## Figures and Tables

**Figure 1 animals-12-03075-f001:**
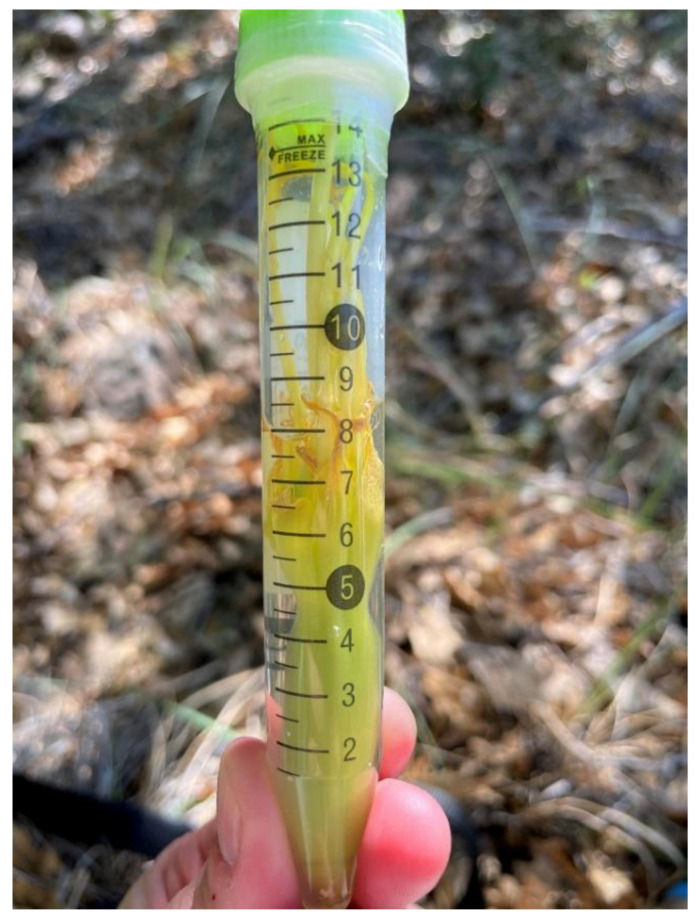
An agave flower placed into a 15 mL conical of RNAlater.

**Figure 2 animals-12-03075-f002:**
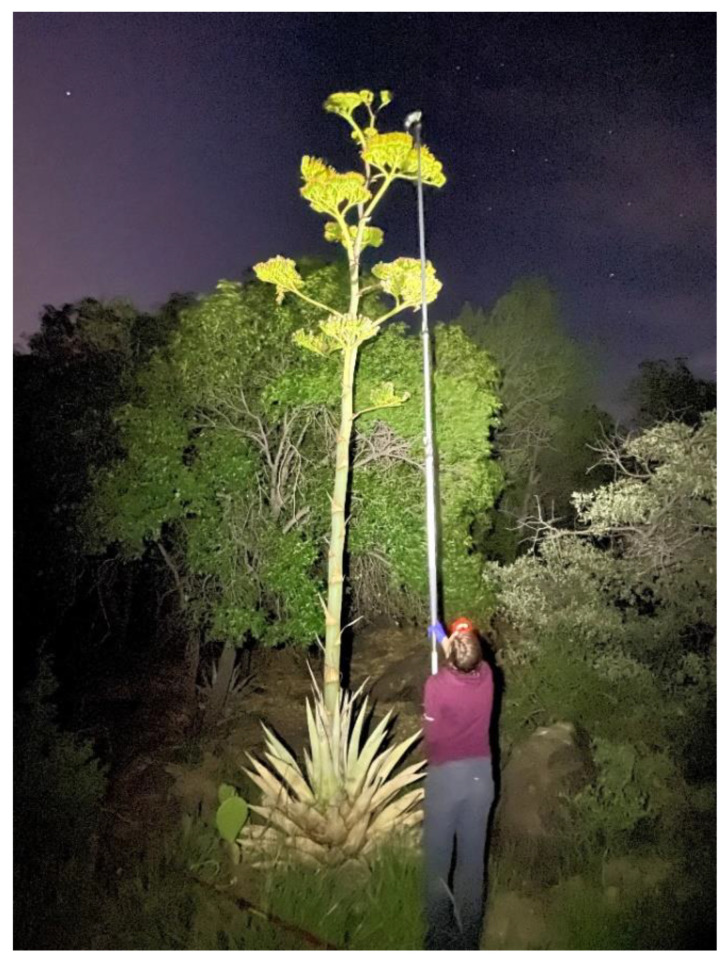
Swabbing agave flowers visited by *L. nivalis* using a telescoping pole with a swab affixed to the top.

**Table 1 animals-12-03075-t001:** Metabarcoding primers (uppercase) used in multiplex PCR for detection of *L. nivalis* (or other vertebrates), plants, and arthropods from agave flower cuttings or swabs. Lowercase nucleotides are 5′-modified universal tails [34] that allowed for Illumina library preparation in a second PCR step.

Primer Name	Primer Sequence (5’–3’)	Target Region	Target Taxa	Insert Size (bp)	Source
SFF145f (forward)	acccaactgaatggagcGTHACHGCYCAYGCHTTYGTAATAAT	COI	Bats	202	[30]
SFF351r (reverse)	acgcacttgacttgtcttcCTCCWGCRTGDGCWAGRTTTCC				
ITS2 (forward)	acccaactgaatggagcATGCGATACTTGGTGTGAAT	ITS2	Plants	100–480	[32]
ITS3 (reverse)	acgcacttgacttgtcttcGACGCTTCTCCAGACTACAAT				
rbcl2 (forward)	acccaactgaatggagcTGGCAGCATTYCGAGTAACTC	rbcL	Plants	96	[33]
rbclA (reverse)	acgcacttgacttgtcttcCCTTTRTAACGATCAAGRC				
ANML-LCO1490 (forward)	acccaactgaatggagcGGTCAACAAATCATAAAGATATTGG	COI	Arthropods	~185	[31]
ANML-CO1-CFMRa (reverse)	acgcacttgacttgtcttcGGWACTAATCAATTTCCAAATCC				

**Table 2 animals-12-03075-t002:** Candidate qPCR assays (forward primer, reverse primer, and hydrolysis probe) developed and used for detection of *L. nivalis*. Assay-specific information also includes optimal primer and probe concentrations, thermocycling conditions, and mean number of mismatches to a closely related congener, *L. yerbabuenae* (LEYE).

Assay Name		Sequence (5′ to 3′)	Tm (°C)	Amplicon Length (bp)	Optimal Conc. (nM)	Cycling Conditions	Mismatch (LEYE)
LENIS1	Forward	CATACTCCACACGTCCAAAC	51.8	156	900	95 °C for 15 s	3
	Reverse	TAGGATGGATGCTACCTGTC	51.8		900	54 °C for 30 s	4
	Probe	FAM-AGGGATGTTCGACTGGTTGGCCTC-ZEN/Iowa Black™ FQ	60.8		250	72 °C for 30 s	3
LENIS3	Forward	TTGTAGCGACCCTGCTTAC	51.1	88	600	95 °C for 15 s	6
	Reverse	TAGGATGGATGCTACCTGTC	51.8		300	60 °C for 30 s	4
	Probe	FAM-AGGGATGTTCGACTGGTTGGCCTC-ZEN/Iowa Black™ FQ	60.8		250		3

**Table 3 animals-12-03075-t003:** *L. nivalis* detections from flower samples (*n* = 13) of two agave plants, with various sampling, stabilization, extraction methods (number detected|number tested), substrates, or DNA extraction kits. Cut1 consists of stigma, style, and ovary, shipped un-stabilized. Cut2 consists of stigma, style, and ovary, shipped in RNAlater. Cut3 consists of half of a pistil and remaining RNAlater. DBT = Dneasy Blood and Tissue Kit. DPT = Dneasy Plant Tissue kit. Samples marked as “n/a” were untested for a detection method. LS = Sierra Madre Oriental in Laguna de Sanchez, Nuevo León, Mexico; CB = Chisos Mountains in Big Bend National Park, Texas, USA.

	Metabarcoding (SFF COI)		qPCR	
Agave Plant	Multiplex	Singleplex	qPCR.LENIS1	qPCR.LENIS3
CB: Cut1—DBT	2|3	2|3	2|3	2|3
CB: Cut2—DBT	1|3	0|3	1|3	2|3
CB: Cut3—DBT	3|3	2|3	2|3	2|3
CB: Swab—DBT	3|3	3|3	3|3	3|3
LS: Cut1—DBT	3|4	n/a	4|4	4|4
LS: Cut1—DPT	1|4	n/a	1|4	1|4
CB (all)	8|9	7|9	7|9	8|9
LS (all)	3|4	n/a	4|4	4|4
Total	11|13	7|9	11|13	12|13

**Table 4 animals-12-03075-t004:** Standard curve (six-level; fourfold) regression coefficients, quantification reliability (efficiency and LOQ), and sensitivity estimates (LOD via probit modeling). Estimates of LOD and LOQ reflect copy numbers per reaction. Copy numbers have been rounded up to the nearest integer.

Metric	LENIS1	LENIS3
R^2^	0.99	0.98
Slope	−3.39	−3.59
Y-intercept	42.52	42.38
Efficiency (%)	97.07	90.07
LOQ	174	462
LOD	57	84
LOD: 2 replicates	25	30
LOD: 3 replicates	15	20
LOD: 4 replicates	11	16
LOD: 5 replicates	8	13
LOD: 8 replicates	5	9

**Table 5 animals-12-03075-t005:** Taxonomies derived from DNA metabarcoding via multiplex PCR of agave flowers visited by *L. nivalis*. Taxa observed more than once among biological and technical replicates of a plant are noted by **. All taxa in grey have been validated as occurring in the study area. BioSample accession numbers are in Appendix A.

Taxonomic Group	Order	Family	Species	Common Name
Bats	Chiroptera	Phyllostomidae	*Leptonycteris nivalis* **	Mexican long-nosed bat
Non-target vertebrates	Artiodactyla	Cervidae	*Odocoileus hemionus or virginianus* **	Carmen whitetail deer or mule deer
	Carnivora	Canidae	*Urocyon cinereoargenteu* **	gray fox
	Carnivora	Procyonidae	*Bassariscus astutus*	ringtail cat
Plants	Asparagales	Agavaceae	*Agave* sp. **	agave
	Gentianales	Rubiaceae	*Bouvardia ternifolia* **	firecracker bush
	Myrtales	Onagraceae	*Onagraceae* sp. **	Evening Primrose family
	Poales	Poaceae	*Sporobolus airoides* **	alkali sacaton
	Poales	Poaceae	*Bouteloua curtipendula*	sideoats grama
	Lamiales	Oleaceae	*Menodora* sp.	menodora
	Poales	Poaceae	*Muhlenbergia* sp.	muhly
	Asterales	Asteraceae	*Parthenium* sp.	feverfew
	Lamiales	Plantaginaceae	*Plantaginaceae* sp.	Plantain family
	Sapindales	Anacardiaceae	*Rhus virens*	evergreen sumac
	Gentianales	Rubiaceae	*Rubiaceae* sp.	madder family
	Lamiales	Lamiaceae	*Salvia* sp.	sage
	Malvales	Malvaceae	*Sphaeralcea* sp.	globemallow
	Lamiales	Bignoniaceae	*Tecoma stans*	yellow trumpetbush
	Malpighiales	Euphorbiaceae	*Tragia* sp.	noseburn
Arthropods	Hymenoptera	Apidae	*Apis* sp. **	honey bee
	Diptera	Cecidomyiidae	*Cecidomyiidae* sp. **	gall and wood midge
	Coleoptera	Cerambycidae	*Cerambycidae* sp. **	longhorn beetle
	Diptera	Ceratopogonidae	*Ceratopogonidae* sp. **	biting midge
	Coleoptera	Cleridae	*Cleridae* sp. **	checkered beetle
	Coleoptera		*Coleoptera* sp. **	beetle order
	Trombidiformes	Eupodidae	*Eupodidae* sp. **	prostig mite
	Thysanoptera	Thripidae	*Frankliniella occidentalis* **	thrip
	Hymenoptera	Colletidae	*Hylaeus* sp. **	masked bee
	Entomobryomorpha	Isotomidae	*Isotomidae* sp. **	elongate-bodied springtail
	Hymenoptera	Halictidae	*Lasioglossum jubatum* **	sweat bee
	Lepidoptera		*Lepidoptera* sp. **	butterfly and moth order
	Hemiptera	Coreidae	*Leptoglossus zonatus* **	leaf-footed bug
	Lepidoptera	Erebidae	*Melipotis indomita* **	underwing moth
	Lepidoptera	Apatelodidae	*Olceclostera seraphica* **	seraph moth
	Coleoptera	Tenebrionidae	*Tenebrionidae* sp. **	darkling beetle
	Araneae	Thomisidae	*Thomisidae* sp. **	crab spider
	Orthoptera	Romaleidae	*Brachystola magna*	grassland lubber
	Hemiptera	Pentatomidae	*Brochymena hoppingi*	rough stink bug
	Diptera	Chloropidae	*Chloropidae* sp.	fruit fly
	Diptera	Syrphidae	*Copestylum* sp.	hoverfly
	Diptera	Culicidae	*Culex pipiens*	house mosquito
	Lepidoptera	Notodontidae	*Datana* sp.	prominent moth
	Phasmatodea	Diapheromeridae	*Diapheromeridae* sp.	walkingstick
	Diptera		*Diptera* sp.	fly order
	Entomobryomorpha	Entomobryidae	*Entomobryidae* sp.	slender springtail
	Lepidoptera	Pyralidae	*Ephestiodes gilvescentella*	pyralid moth
	Lepidoptera	Erebidae	*Erebidae* sp.	Erebidae moth
	Hymenoptera		*Hymenoptera* sp.	ant, bee, wasp, sawfly order
	Hymenoptera	Halictidae	*Lasioglossum* sp.	sweat bee
	Uropygi	Thelyphonidae	*Mastigoproctus giganteus*	giant vinegarroon
	Lepidoptera	Erebidae	*Matigramma emmilta*	owlet moth
	Orthoptera	Acrididae	*Melanoplus walshii*	Walsh’s short-wing grasshopper
	Mesostigmata	Melicharidae	*Melicharidae* sp.	melicharid mite
	Hemiptera	Membracidae	*Membracidae* sp.	typical treehopper
	Hemiptera	Aphididae	*Myzus persicae*	green peach aphid
	Coleoptera	Nitidulidae	*Nitops pallipennis*	Nitops sap-feeding beetle
	Diptera	Tachinidae	*Prorhynchops* sp.	Tachinid fly
	Hemiptera	Miridae	*Rhinacloa forticornis*	western plant bug
	Sarcoptiformes	Scheloribatidae	*Scheloribatidae* sp.	Acari mite
	Mantodea	Mantidae	*Stagmomantis californica*	California mantis
	Orthoptera	Tettigoniidae	*Tettigoniidae* sp.	katydid

## Data Availability

Sequencing data were deposited in NCBI under BioProject ID PRJNA890273. BioSample accessions are in Table 2.

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
