# Peer review of "Endangered Nectar-Feeding Bat Detected by Environmental DNA on Flowers"

_animals, 2022, doi:10.3390/ani12223075_

Round 1

Reviewer 1 Report

This is a very interesting research on the detection of endangered nectar-feeding bat by environmental DNA on flowers. The manuscript well written, introduction gives a comprehensive explanation of the essence of the article, materials and methods well described, results clearly presented. In conclusions, supported by the results the authors propose a research on a larger scale, which is very much promising. I suggest to accept this article  after the text editing. I enclosed pdf. file with some suggestion.

Reviewer 2 Report

I checked your manuscript and described comments below.

This paper provides a very good analysis of the relationship between Leptonycteris nivalis and the “nectar corridors” using eDNA.

I think ref. 21 and 22 are better for Journal than for PhD dissertation.

I don't think this paper has any major mistakes or grammatical problems.

Reviewer 3 Report

This manuscript is about the development of qPCR and metabarcoding platform for the ecological study the endangered nectar-feeding bat, Leptonycteris nivalis, which has been difficult by the traditional methodology. Both qPCR and metabarcoding analyses are among the hot topic for the ecological studies and this application would be one of the beneficial and promising tools for the rarely detected species. They followed the typical methods, which are widely used for ecologists and this strategy was successful to detect the endangered bat species. Therefore, this manuscript is worth to be published for those who would like to study migratory avian/chiropteran species. However, my main concern about this manuscript is about the lack of main object in this research. Since this paper is not the ecological interpretation for the bat ecology, author should emphasize the significance, importance, and purpose of this manuscript for readers, for instance, establishment of a standardized method for the bat species or feasibility examination of the used universal primers. Therefore, I would like to authors to rearrange the manuscript focusing two main objects commented above.

First, I am afraid the cutting the pollen stems are still the destructed method for the study. Using one of table or figure, It would be nice to show potential problem in standardizing sample collection method and preparation of DNA for the bat eDNA study. Below is one example of standardization of eDNA montoring.

 https://www.csagroup.org/article/research/environmental-dna-standardization-needs-for-fish-and-wildlife-population-assessments-and-monitoring/

2. Comparative analysis of universal primers targeting both plants and arthropods. I understand authors thoroughly studies the feasibility and reliability of the used universal taxon primers and the primer sets used in this study are among the best ones. However, In order to persuade the readers, I strongly recommend to compare ngs results using another widely used universal primer set. For instance, two primers for plant (https://www.nature.com/articles/s41598-018-26648-2) and

3. All the tables with species name should contain GenBank numbers and sequence similarity (%) to the reference sequence data.

Minor.

Line 285 In order to prove the feasibility of the primer set, it is required to authors to provide any experimental data, such as multiple alignment in supplementary. Besides, is there any intraspecies sequence variations within the amplicon region?

Table 2. L. nivalis should be italic. Please it throughout the manuscript.
